# High-Fat Diet Impairs Muscle Function and Increases the Risk of Environmental Heatstroke in Mice

**DOI:** 10.3390/ijms23095286

**Published:** 2022-05-09

**Authors:** Matteo Serano, Cecilia Paolini, Antonio Michelucci, Laura Pietrangelo, Flavia A. Guarnier, Feliciano Protasi

**Affiliations:** 1CAST, Center for Advanced Studies and Technology, University G. D’Annunzio of Chieti-Pescara, 66100 Chieti, Italy; matteo.serano@unich.it (M.S.); cecilia.paolini@unich.it (C.P.); antonio.michelucci@unich.it (A.M.); laura.pietrangelo@unich.it (L.P.); 2DMSI, Department of Medicine and Aging Sciences, University G. D’Annunzio of Chieti-Pescara, 66100 Chieti, Italy; 3DNICS, Department of Neuroscience and Clinical Sciences, University G. D’Annunzio of Chieti-Pescara, 66100 Chieti, Italy; 4Department of General Pathology, Londrina State University, Londrina 86057-970, Brazil; faguarnier@uel.br

**Keywords:** heat stroke, high-fat diet, malignant hyperthermia susceptibility, skeletal muscle

## Abstract

Environmental heat-stroke (HS) is a life-threatening response often triggered by hot and humid weather. Several lines of evidence indicate that HS is caused by excessive heat production in skeletal muscle, which in turn is the result of abnormal Ca^2+^ leak from the sarcoplasmic reticulum (SR) and excessive production of oxidative species of oxygen and nitrogen. As a high fat diet is known to increase oxidative stress, the objective of the present study was to investigate the effects of 3 months of high-fat diet (HFD) on the HS susceptibility of wild type (WT) mice. HS susceptibility was tested in an environmental chamber where 4 months old WT mice were exposed to heat stress (41 °C for 1 h). In comparison with mice fed with a regular diet, mice fed with HFD showed: (a) increased body weight and accumulation of adipose tissue; (b) elevated oxidative stress in skeletal muscles; (c) increased heat generation and oxygen consumption during exposure to heat stress; and finally, (d) enhanced sensitivity to both temperature and caffeine of isolated muscles during in-vitro contracture test. These data (a) suggest that HFD predisposes WT mice to heat stress and (b) could have implications for guidelines regarding food intake during periods of intense environmental heat.

## 1. Introduction

Global warming has become a reason for concern for human health due to the dramatic rise in mortality rate during heat waves [1,2]. Heat waves are periods of abnormally and uncomfortably hot and usually humid weather [3]: according to the American Meteorological Society glossary, a heat wave corresponds to a period of three consecutive days during which the maximum temperature is above the threshold of 32.2 °C.

The impact of heat waves on human health, especially in urban areas, is very well documented in literature (hundreds of scientific reports) and, surprisingly, these reports indicate that more than 90% of human deaths in natural hazards are caused by hot weather [4]. Some specific heat waves are very well documented: the 2003 France heat wave was accompanied by an excess mortality of exceptional magnitude: 14,947 excess deaths in the period of 4–18 August [5]. Interestingly mortality returned to its normal level starting on 19 August. Just a few months ago, dozens of people died in Canada in an unprecedented heat wave that smashed temperature records (close to 50 °C). These are only a few examples of a very long list [6,7]. Even if several factors may contribute to sudden death in high environmental temperatures, the most common cause of death attributable to heat is dehydration, heat cramps and exhaustion, and hyperthermia, i.e., in one word heat stroke (HS) [8]. HS is a life-threatening response caused by exposure to environmental heat often combined to high humidity (environmental HS). HS is characterized by a rapid increase in core body temperature (above 40 °C), culminating in dysfunction of organs and of the central nervous system [9].

However, while the correlation between environmental heat and increased mortality is now clear, the mechanisms mediating multi-organ damage, the drugs to be used in emergency situations, and finally life habits that may prevent or reduce triggering of HS in hot weather are still debated.

In humans, there is a syndrome characterized by an abnormal increase in body temperature known as malignant hyperthermia susceptibility (MHS). MHS is a genetic life-threatening pharmacogenetic response triggered by halogenated anesthetics such as isoflurane or halothane [10,11,12], which shares many pathophysiological features with HS: rhabdomyolysis, increase in serum creatine kinase, hyperkalemia, tachycardia, metabolic acidosis, and unbridled rise in body temperature [9,13]. Patients who have experienced heat stress, which is the perceived discomfort that precedes HS have higher incidence of being positive in-vitro contracture tests (IVCT), the gold-standard test used to determine MHS of patients. IVCT measures the contractile sensitivity of a muscle biopsy to triggering agents such as caffeine and halothane [14,15,16].

Correlation between MHS and HS (hypothesized in humans) has been already demonstrated in animal models carrying specific gene mutations: (a) swine that carry a point mutation in the skeletal muscle ryanodine receptor type-1 (RyR1) trigger MH episodes in response to halothane administration, but also during exposure to either heat or emotional/physical stress [17,18]; (b) knock-in mice that carry gain-of-function mutations in the RyR1 gene that are causative of MHS in humans (R163C and Y522S) are susceptible to lethal overheating crises when exposed to either halogenated anesthetics or elevated temperature [19,20]; and finally (c) mice that lack the protein calsequestrin-1 (CASQ1-null) also exhibit lethal anesthetic- and heat-induced hyperthermic episodes [21,22,23,24]. Studies from our group and others indicates that rhabdomyolysis of skeletal muscle fibers during MHS/HS crisis is the final result of a complex cascade of events starting from excessive Ca^2+^ leak from RyR1 mediated by oxidative stress and consequent excessive nitrosylation of RyR1 [25]. In support of this view, we demonstrated: (a) that antioxidants protect CASQ1-null mice from anesthetic- and heat-induced lethal crisis; and (b) how moderate aerobic training and administration of estrogens reduced the mortality rate of CASQ1-null mice during heat stress reducing oxidative stress and mitochondrial damage [22,26].

While correlation between MHS and HS has been demonstrated in animals carrying specific gene mutations, which factors and how lifestyle habits may either increase or diminish the risk of the normal population to HS in challenging environmental conditions remain to be determined. According to the WHO (World Health Organization), the prevalence of obesity nearly tripled between 1975 and 2016. The fundamental cause of obesity and overweight is the energy imbalance between calories consumed and calories expended, meaning increased intake of energy-dense foods that are high in fat and sugars combined with reduced physical inactivity. Interestingly, excessive consumption of calories, especially those related to lipids and fatty acids, has been associated to increased inflammatory biomarkers and oxidative stress levels [27]. A diet rich in fat may induce oxidative stress, potentially through the up-regulated expression of genes for ROS production and through mitochondrial respiratory chain overload [28]. 

As mice models susceptible to HS crises display high levels of oxidative stress [22,23,25,26,29], in the present study, we tested whether a high-fat diet administrated to wild type (WT) mice may predispose to environmental HS crisis. 

## 2. Results

### 2.1. High-Fat Diet (HFD) Induced Metabolic Modifications in Adult WT Mice

We monitored food intake, body weight gain, and evaluated the percentage of adipose tissue in control (Ctrl) and HFD mice during the 3 months of diets (Figure 1). No significant differences were found between the amount of chow consumed by the two experimental groups of mice (Figure 1A). However, even though the amount of food ingested was approximately the same, the HFD group of mice showed an increase in body weight (15.65 ± 0.44 gr) much greater than the Ctrl group (4.73 ± 0.94 gr, *p* < 0.01) (Figure 1B). 

We also quantified the amount of body fat dissecting the adipose tissue from each animal, as shown in Figure 1C,D: this analysis indicated that most of the excessive weight in HFD mice was accumulated in body fat: 11.28 ± 0.63% in Ctrl vs. 20.68 ± 1.30% in HFD (Figure 1E). 

We then used indirect calorimetry to evaluate whether the HFD diet modified the oxygen consumption and the metabolic rate of mice (Figure 2). 

The results shown in Figure 2 indicate that HFD induced a significant reduction in VO_2_ consumption at rest, compared to Ctrl: on the average a decrease of 25% of VO_2_ consumption in both light and dark cycles (Figure 2A). In line with these results, the respiratory quotient (RQ = VCO_2_ produced/VO_2_ consumed) was also different. Figure 2B shows that Ctrl mice, at rest, maintain their RQ curve between 0.8 and 0.9 in both light and dark cycles. On the other hand, the consumption of HFD mice over 3 months switched the RQ to constant values very close to 0.75, which indicates a predominantly lipid-oriented metabolism compared to Ctrl. The results above mentioned were reinforced by average basal energy expenditure (BEE) (Figure 2C), showing an average decrease of 20% in HFD mice during all circadian cycle when compared to Ctrl.

### 2.2. HFD Caused Muscle Dysfunction

In-vivo muscle function was evaluated measuring grip strength: HFD mice displayed a significant reduction in normalized grip strength (mN/g) (~31% of reduction compared to Ctrl) (Figure 3A).

We then determined the functional output of isolated muscles, evaluating the force/frequency response of isolated muscles in an ex-vivo set up. EDL muscles dissected from HFD mice showed a reduction in specific force curve starting from 75 Hz (~20% of reduction) (Figure 3B). To determine if the difference in specific force was caused by a switch in fiber type composition, we normalized the relative force to 250 Hz (Figure 3C). As no statistical difference was detected between the two groups, the possibility of a switch in fiber type was ruled out (Figure 3C). Interestingly, we found comparable results even in Soleus: (a) muscles from HFD mice showed significant reduction in specific force starting at ~50 Hz (~14% of reduction; Appendix A); (b) also in this case, when relative force was normalized to 200 Hz, no statistical difference was found between Ctrl and HFD (Appendix A). 

### 2.3. Oxidative Stress Was Elevated in Muscles from HFD Mice

As excessive production of reactive oxygen and nitrogen species (ROS and RNS) has been proposed to be a key step in the cascade of molecular events that leads to HS crises in MH susceptible mice [21,23,25,26,30], we investigated the presence of markers of oxidative stress and damage in EDL and Soleus homogenates from our two experimental groups. First, we assayed by Western blot (WB) the amount of 3-nitrotyrosine (3-NT) (Figure 4A), a product of nitration of tyrosine residues of proteins mediated by RNS [31]. EDL from HFD mice contained levels of 3-NT 60% higher when compared to Ctrl. In addition, we evaluated levels of: (i) copper/zinc superoxide dismutase (SOD-1) (Figure 4B) and manganese superoxide dismutase (SOD-2) (Figure 4C), the two main intracellular isoforms of a class of enzymes that catalyze the dismutation of O_2_^−^ into O_2_ and H_2_O_2_, the first step in the scavenging of ROS generated primarily in mitochondria [32]; and (ii) Catalase (Figure 4D), one of the most important antioxidant enzymes that catalyzes the decomposition of H_2_O_2_ to H_2_O and O_2_ and that represents an important antioxidant defense for skeletal muscle, acting immediately after the action of SOD-1 and SOD-2 [33]. In EDL muscles from HFD mice, the level of these enzymes was increased respectively by ~28% (SOD-1), ~31% (SOD-2), and ~167% (Catalase) when compared to Ctrl (Figure 4B–D).

Similar results were collected in the Soleus muscle, a predominantly slow twitch muscle (Appendix A): 2.0-fold increasing in 3-NT (Appendix A), and ~40% (Appendix A) and ~26% (Appendix A) increase in SOD-1 and SOD-2, respectively. Finally, Catalase also showed increased levels in the HFD group: ~41% when compared with Ctrl (Appendix A). 

### 2.4. HFD Induced Increased Metabolic Rate during Heat Stress

We exposed mice to a heat stress protocol (41 °C for 1 h) using a custom-made environmental chamber in which we could control temperature and humidity. While exposed to heat stress, mice were kept inside metabolic chambers to evaluate VO_2_ consumption and basal energy expenditure (BEE) by indirect calorimetry.

The HFD significantly increased the VO_2_ consumed during heat stress (Figure 5A): the difference between initial (T_0_) and final (T_f_) values of VO_2_ consumption reached 5 mL/min/kg^0.75^ in Ctrl, while in HFD mice this difference was about 7 mL/min/kg^0.75^ in some mice. Likewise, under heat stress, energy expenditure of HFD mice increased by ~13% when compared to Ctrl (Figure 5B). To verify if the increased metabolism was accompanied by changes in body temperature, we registered the body and skin temperature of each mouse before and after the heat stress protocol. Significant differences were found both in core (Figure 5C) and skin (Figure 5D) temperatures after the heat stress protocol. Core temperature, assayed with a rectal thermometer, increased both in Ctrl and HFD mice, but more in the latter: 35.38 ± 0.06 °C to 39.20 ± 0.13 °C in Ctrl (ΔT = +3.82); 35.97 ± 0.38 °C to 40.53 ± 0.26 °C in HFD (ΔT = +4.56). Measurements were also performed using an infrared thermometer: skin temperature was higher in HFD mice already before the heat stress protocol (31.50 ± 0.24 °C in Ctrl vs. 32.13 ± 0.14 °C in HFD), increasing during the heat stress protocol to 33.43 ± 0.44 °C vs. 35.12 ± 0.45 °C in Ctrl (ΔT = +1.93) and HFD (ΔT = +2.99), respectively.

### 2.5. HFD Induced Hypersensitivity of Muscles to IVCT and Increased Levels of Markers of Muscle Damage

In-vitro contracture test (IVCT) is the gold standard technique to evaluate the oversensitivity of muscles to heat [34,35]. We performed an in-vitro heat stress protocol, based on exposure of isolated muscles to increasing steps of temperature of 2 °C each (Figure 6A). 

When exposed to this protocol, EDL muscles excised from HFD mice showed a significant increase in basal tension starting from ~38 °C (Figure 6A), that at 44 °C became ~30% higher than in Ctrl. We performed similar experiment in Soleus muscles, also detecting a difference in heat sensitivity: increase in basal tension compared to Ctrl starting at ~40 °C (Appendix A), with a relative basal tension at the end of the experiment (44 °C) which was ~15% higher in the Soleus from HFD compared to Ctrl.

We then performed a classic caffeine dose-response experiment, mimicking the IVCT that is used in human biopsies to test MH susceptibility [34,35]. Caffeine is a potent agonist of RyR1 that triggers release of Ca^2+^ from the sarcoplasmic reticulum (SR): MH susceptible patients usually display a lower threshold of response to caffeine [36,37]. In Figure 6B, the basal tension in EDL muscles of HFD mice starts to be significantly higher than Ctrl starting at 18 mM of caffeine. At the end of the experiments, basal tension at 26 mM of caffeine was higher in HFD of ~14%, compared to that recorded in Ctrl. 

Environmental HS may be accompanied by muscle damage and rhabdomyolysis, which can be assessed by blood and histological analyses. In mice exposed to the heat stress protocol (41 °C for 1 h) (Figure 7): plasma levels of K^+^ (7.51 ± 0.32 mmol/L; Figure 7A), Ca^2+^ (1.60 ± 0.09 nmol/L; Figure 7B) and total creatine kinase (CK) (161.26 ± 20.04 U/L; Figure 7C) from HFD were all significantly increased in comparison to values in Ctrl samples (6.31 ± 0.33 mmol/L; 1.35 ± 0.05 nmol/L; and 99.42 ± 16.00 U/L, respectively).

We finally quantified, in histological sections, the percentage of EDL fibers that were affected by structural damage after heat stress (Appendix A). EDL fibers were classified into three main groups that presented the following typical features: (i) fibers with no apparent damage (Appendix A); (ii) fibers with unstructured cores (Appendix A); and (iii) fibers with contracture cores (Appendix A). No detectable differences were registered between the two experimental groups (Appendix A). 

## 3. Discussion

### 3.1. State of the Art

In the latest years, global warming has become a reason for concern for human health due to the dramatic rise in mortality rate during heat waves [2,38]. An increase in mortality in these periods has been associated to cardiovascular, respiratory, and kidney diseases [39,40,41,42], to nervous systems dysfunction [43,44,45], and also to diabetes [42,46]. 

Even if several factors may contribute to death in high environmental temperatures (age, pre-existing disease, urban residence, isolation, poverty, air pollution), the most common cause of death attributable to heat is dehydration, heat cramps, exhaustion, and hyperthermia, i.e., in one word, heat stroke (HS) [9]. Our group has contributed in the last decade to studying the molecular mechanisms underlying HS: excessive Ca^2+^ leak from the SR and overproduction of oxidative stress in skeletal muscle fibers seem to play an important role in these events [25]. We hypothesized that HFD may be a factor of risk for HS, because consumption of excessive amount of fats has already been associated to increase in oxidative stress [28]. 

### 3.2. Main Findings

The results collected in the present study indicated that 3 months of HFD was able to produce anatomical changes: increase in body weight and in total adipose tissue mass, though, without significant modifications in the amount of food ingested (Figure 1). These changes were accompanied by a decrease in the level of energy spent for basal metabolism when compared to Ctrl (Figure 2), by muscle dysfunction (Figure 3 and Appendix A) and, finally, by increased levels of oxidative stress in muscle, both in fast and slow twitch (Figure 4 and Appendix A). While BEE was reduced in standard housing conditions (Figure 2), during heat stress, mice fed with HFD showed a significantly increased VO_2_ consumption and increased body temperature (Figure 5). These changes were accompanied by isolated muscle being more prone to develop contracture during IVCT in response to both temperature and caffeine (Figure 6 and Appendix A) and by increased circulating levels of markers of muscle damage (Figure 7).

### 3.3. Changes in Basal Metabolism, Oxidative Stress, and Muscle Dysfunction Caused by HFD

Increase in body weight and increase in adipose tissue in mice fed HFD were accompanied by significant modifications in metabolism: lower metabolic rate and a shift toward the use of lipids, when compared to Ctrl, as shown by the decreased RQ (Figure 2). RQ gives information about which is the main substrate oxidized during mitochondrial respiration, i.e., the relative amount of glucose, lipids, or proteins used to produce energy [47]. RQ is different for carbohydrate, lipid, and proteins, because the VCO_2_/VO_2_ ratio is higher in proteins and glucose than in lipids: a ratio between 0.7 and 0.8 indicates the prevalent use of lipids and proteins as substrates, whereas a ratio above 0.8 indicates increasing use of carbohydrates as the main source of energy. The consumption of HFD mice switched to RQ values very close to 0.75, significantly lower than controls (Figure 2). Another parameter that was evaluated is the BEE, which indicates the energy required for the normal functioning of cells and organs. It is calculated based on VO_2_ consumption registered by indirect calorimetry and it is proportional to it, both values being representative of total metabolism [48]. HFD administration caused a decrease in both VO_2_ consumption and BEE sufficient to lower total metabolic rate at rest of about ~25% compared to Ctrl mice (Figure 2). This is consistent with data published in humans showing a decreased basal energy expenditure in individuals chronically fed a diet containing 40% more fats compared to a control diet [48,49], although some genetic individual variability was found [50]. In our mice, the metabolic changes caused by HFD were accompanied by oxidative stress and dysfunction of skeletal muscle, the organ responsible for the overgeneration of heat during heat stress. Levels of oxidative stress were significantly elevated both in fast and slow twitch muscle in HFD mice when compared to Ctrl (Figure 4 and Appendix A). In the past 15 years, we collected strong evidence that oxidative stress is the key event underlying the increased susceptibility to heat, at least in animal models. The mechanistic hypothesis is that excessive production of ROS and RNS during exposure to heat would cause a dangerous feed-forward cycle that involves excessive S-nitrosylation of RyR1 (i.e., the Ca^2+^ release channel of the SR), which increases its opening probability [23,24,25,30]. The consequent excessive release of Ca^2+^ from the SR would finally promote the contracture and rhabdomyolysis of skeletal muscle fibers that underlie MHS/HS reactions. 

Finally, skeletal muscle dysfunction was assessed by in vivo and in vitro experiments. Decreased values in grip strength test and reduction in force production both in EDL and Soleus muscle when subjected to in vitro stimulation protocols were detected (Figure 3 and Appendix A). The decrease in force may be caused by several factors, one of them possibly being oxidative stress itself, as ROS and NOS are known to cause oxidative damage to proteins and membranes and to alter their function [33,51]. For instance, aging is indeed a physiological condition in which oxidative stress is elevated and accompanied by reduced force output [52,53,54,55,56].

### 3.4. Increased Susceptibility to Heat Stress in HFD Mice

While in normal housing conditions, the basal metabolism of HFD mice was reduced compared to Ctrl, when we subjected mice to environmental heat stress (1 h at 41 °C), both BEE and VO_2_ consumption were surprisingly increased in HFD compared to Ctrl (Figure 5). Increased metabolism was also accompanied by increase in core and skin temperatures during the heat stress protocol (Figure 5). The greater increase in body temperature could be the result of two factors: (1) the increase in oxygen consumption and metabolism and (2) the increased amount of adipose tissue that, being an insulating tissue, could limit the dissipation of heat through the skin. 

The classic clinical features of MHS and HS crises are: hypermetabolism, increased internal temperature, increased circulating levels of K^+^, and creatine kinase (CK), and finally rhabdomyolysis [9,14]. Here, in addition to increased metabolism and excessive rise in body temperature (discussed above; Figure 5), we also measured circulating factors that may suggest muscle damage detecting increased levels of K^+^, Ca^2+^, and CK (Figure 7). Histological analyses, though, revealed no significant difference in the percentage of damaged EDL fibers between Ctrl and HFD mice (Appendix A). The increase of blood markers of muscle damage without a visible change in histology suggests that damage, where present, is still mild. 

Finally, because muscles from MHS mice display high susceptibility to develop contracture when isolated muscles are subjected to increasing concentration of caffeine and heat [21,25], here, we also tested muscles with IVCT: EDL, as well as Soleus, muscles from HFD mice indeed developed higher basal tension compared to Ctrl in response to both increasing temperature and caffeine (Figure 6 and Appendix A). Increased generation of tension indicate higher levels of myoplasmic Ca^2+^ concentration, which is indicative of SR Ca^2+^ leak through RyR1, and a recognized hallmark of MHS/HS crises.

## 4. Materials and Methods

### 4.1. Animals and Experimental Design

All experiments were conducted according to the Directive of the European Union and were approved by the Animal Ethical Committee of the University of Chieti-Pescara. Male C57bl/6 mice of 1 month of age were randomly assigned to two experimental groups: Control (Ctrl), fed with a regular diet and high-fat diet (HFD) group. In the control diet, 10% of metabolized energy was from fats (C 1090–10, Altromin, Lage, Germany), whereas HFD mice were fed with an obesity-inducing diet (in which 70% of metabolized energy derived from fats) (C 1090–70, Altromin, Lage, Germany). The complete composition of both Ctrl and HFD diets is detailed in Appendix A. Once assigned to the two experimental groups, mice were housed in microisolator cages at 20 °C in a 12-h light/dark cycle and provided free access exclusively to the respective diets and water (2 animals/cage). Food intake was registered three times a week. The weight of mice was registered at the beginning and at the end of the diet treatment (at 1 and 4 months of age). 

Before all ex vivo experiments, animals were killed by cervical dislocation as approved by the local University Committee on Animal Resources (1202/2020-PR).

### 4.2. In Vivo Experiments

#### 4.2.1. Grip Strength Test

Grip strength of both Ctrl and HFD groups was measured holding mice by the tail and lowering them to a metal grating connected to the shaft of a Shimpo Fgv 0.5X force transducer (Metrotec Group, Lezo, Spain). Once the mouse had firmly grabbed the grating, a gentle pull was exerted on the tail [29]. Measurements of peak force generated by each mouse using fore limbs were repeated three times with appropriate intervals (30 s) to avoid fatigue. Average peak force values were normalized to body weight measured immediately before each experiment.

#### 4.2.2. Indirect Calorimetry

At 4 months of age, mice of both Ctrl and HFD groups were placed individually in sealed home cages intended for measurement of oxygen consumption and allowed to adapt for 24 h. After adaptation, the volume of oxygen consumption (VO_2_) and of carbon dioxide production (VCO_2_) were registered for 24 h using Oxylet-Pro indirect calorimeter system (Pan Lab, Cornellà, Spain). The analysis of the respiratory quotient (RQ) and of the energy expenditure (EE) was performed using the Metabolism software provided with the instrument (Pan Lab, Cornellà, Spain) [57].

#### 4.2.3. Heat Stress Protocol and Core Temperature Recording

In order to determine the sensitivity of mice to heat, Ctrl and HFD mice were subjected to a heat stress protocol the day after the assessment of oxygen consumption by indirect calorimetry (see above). Briefly, mice were first placed individually in the same home cages used for indirect calorimetry experiments and then the same cages were placed inside an environmental chamber where the temperature was kept at 41 °C for 1 h (two mice—one of each experimental group—were tested simultaneously). During the heat stress protocol, the respiratory gas volumes were registered every 6 min for 1 h. Core and skin temperatures were registered just before and after exposure to heat stress test in each mouse using four channels (TM-946, XS instruments, Carpi, Italy) and an infrared thermometer (CW364, RS Components, Milan, Italy), respectively.

### 4.3. Experiments after Animals Euthanasia

#### 4.3.1. In Vitro Contracture Test (IVCT)

Extensor digitorum longus (EDL) muscles were dissected from both experimental groups (Ctrl and HFD) and placed in a dish containing Krebs–Hanseleit solution (NaCl 137 mM, KCl 5 mM, MgSO_4_ 1 mM, CaCl_2_ 2 mM, NaH_2_PO_4_ 1.2 mM, HEPES 0.5 mM, and glucose 10 mM). EDL were then tied at each end with fine silk sutures, and mounted vertically between two platinum electrodes, attached to a servomotor and force transducer (model 1200A, Aurora Scientific, Aurora, ON, Canada). Once mounted, EDL muscles were immersed in a chamber filled with Krebs–Hanseleit solution, and before starting the experimental protocol, stimulation level and optimal muscle length (L0) were determined using a series of 80 Hz-tetani, in order to stretch the muscle to a length that generated maximal force (F0). After determining optimal stimulation conditions, EDL muscles were subjected to a force-frequency protocol based on a series of train pulses of 500 ms duration, from 1 to 250 Hz. Specific force (mN/mm^2^) was calculated by normalizing the absolute force (mN) to the cross-sectional area (CSA/mm^2^) obtained as from the following formula: muscle wet weight (mg)/L0 (mm) × 1.06 (mg/mm^3^). 

In order to determine the caffeine sensitivity of resting tension, muscles were continuously subjected to electrical stimulation (0.2 s at 0.2 Hz, applied every 5 s; duty cycle, 0.04) at 30 °C and exposed to increasing caffeine concentrations every 3 min, as follows: 0, 2, 4, 6, 8, 10, 14, 18, 22, and 26 mM. 

On the other hand, to evaluate the development of contractures induced by increasing temperature, EDL muscles were electrically stimulated with a series of consecutive twitches (1 ms duration, 0.2 Hz for each twitch, applied every 5 s) and exposed to steps of increasing temperature (each of 2 °C), every 5 min from 30 °C to 44 °C. Muscle force was continuously recorded using a dynamic muscle control software and analyzed using dynamic muscle analysis software (Aurora Scientific, Aurora, ON, Canada).

#### 4.3.2. Blood Analysis

Immediately after heat stress, 100 μL of blood was collected from Ctrl and HFD mice’s tail vein, then transferred to 1.5 mL micro tubes, centrifuged at 3000× *g* for 15 min at 4 °C for plasma separation, and stored in 200 μL micro tubes at −20 °C until further use. Levels of potassium (Potassium turbidimetric assay kit, myBioSource, San Diego, CA, USA), calcium (Calcium colorimetric assay kit, Sigma-Aldrich, Burlington, MA, USA) ions, and creatine kinase activity (Creatine kinase activity assay kit, Sigma-Aldrich, Burlington, MA, USA) were quantified in plasma samples, following manufacturer’s instructions and expressed in mmol/L, units/L, and nmol/μL, respectively.

#### 4.3.3. Assessment of Rhabdomyolysis

Immediately after the heat stress protocol, EDL muscles were carefully dissected from Ctrl and HFD mice, fixed at room temperature (RT) with 3.5% glutaraldehyde in 0.1 M NaCaCO buffer (pH 7.4), and kept at 4 °C in fixative until further use. Fixed muscles were then postfixed in 2% OsO_4_ in the same buffer for 2 h, then block-stained in uranyl acetate replacement. After dehydration, specimens were embedded in an epoxy resin (Epon 812; Electron Microscopy Sciences, Hatfield, PA, USA) and embedded as previously described [56]. Semi-thin (800 nm) sections were cut with a Leica Ultracut R Microtome (Leica Microsystems, Wien, Austria) using a Diatome diamond knife (Diatome, Nidau, Switzerland). Sections were stained in a solution containing 1% Toluidine blue O and 1% Sodium Borate Tetra in distilled water for 3 min on a hot plate at 55–60 °C. After washing and drying, sections were mounted with DPX mounting medium for histology (Sigma-Aldrich, Milan, Italy) and viewed by using a Leica DMLB light-microscope (Leica Microsystems, Wien, Austria) connected to a Leica DFC450 camera (Leica Microsystem, Wien, Austria). Muscles fibers were then classified in: (i) apparently normal, (ii) fibers with unstructured cores, and (iii) fibers with contracture cores [29]. Quantification of the three different types of fibers was performed and reported as percentage of all evaluated fibers.

#### 4.3.4. Measurements of Oxidative Stress

EDL muscles were rapidly dissected from Ctrl and HFD mice, and quickly frozen in liquid nitrogen until use. At the appropriate time, muscles were homogenized in RIPA buffer (TRIS HCl 50 mM pH 7.4, Triton-X 1%, Deoxycholate 0.25%, NaCl 150 mM, SDS 3%, EDTA 1 mM, Protease inhibitors 2.5%) using a mechanical homogenizer and then centrifuged for 15 min at 900× *g* at 4 °C. The supernatant was collected and protein concentrations were determined spectrophotometrically using a BCA quantification kit (ThermoFisher scientific, Waltham, MA, USA). Equal amounts of total protein (20 μg) were resolved in 12% (for SOD-1 and SOD-2), or in 10% (for Catalase and 3-NT) sodium dodecyl sulfate polyacrylamide gel electrophoresis and transferred to nitrocellulose membranes. Blots were then blocked with 5% non-fat dry milk (EuroClone, Pero, Italy) in TBS-T 0.1% for 1 h. Membranes were probed with primary antibodies diluted in 5% non-fat dry milk in TBS-T overnight at 4 °C, as follows: anti-SOD-1 antibody (rabbit polyclonal 1:1000, Santa Cruz Biotechnology Inc., Dallas, TX, USA), anti-SOD-2 antibody (rabbit polyclonal 1:1000, Santa Cruz Biotechnology, Inc., Dallas, TX, USA), anti-Catalase antibody (rabbit polyclonal 1:1000, Santa Cruz Biotechnology Inc., Dallas, TX, USA), anti-3-NT antibody (mouse monoclonal, 1:500 Merck Millipore, Milan, Italy), and anti-glyceraldehyde-3-phosphate dehydrogenase antibody (GAPDH) (mouse monoclonal, 1:10,000 OriGene Technologies Inc., Rockville, MD, USA) that was used as housekeeping.

After incubation with primary antibodies, membranes were washed with TBS-T and then incubated with secondary antibodies (1:10,000, horseradish peroxidase–conjugated, Merck Millipore, Burlington, MA, USA) diluted in 5% non-fat dry milk in TBS-T for 1 h at RT and washed in TBS-T. The different proteins were detected by enhanced chemiluminescent liquid (Perkin-Elmer, Waltham, MA, USA), and their amount on the immunoblots were quantified using the imaging system Alliance Mini 4 with the Alliance 1D MAX software (UVItec, Cambridge, UK). Membranes incubated with anti-3-NT antibody were stripped by TRIS/SDS buffer with 2-mercaptoethanol. After blocking, membranes were incubated with anti-GAPDH (mouse monoclonal, 1:10,000 OriGene Technologies Inc., Rockville, MD, USA) for normalization to protein content of each band.

Note: A full list of all chemicals/antibodies used in the work is provided in Appendix A.

### 4.4. Statistical Analysis

Data are shown as mean ± SEM. Differences were considered statistically significant at *p* < 0.05 or *p* < 0.01, where indicated. Unpaired Student’s *t*-tests and two-way ANOVA tests were performed using GraphPad Prism 6 software. The specific statistical test used to compare data presented in each figure is reported in the figure legends.

## 5. Conclusions

As global warming is increasing the frequency of heat waves, which may be dangerous for human health especially in urban areas, a deeper understanding of the mechanisms causing HS is urgent. Our study indicates that 3 months of HFD may predispose WT mice to HS susceptibility, and that HS susceptibility of mice subjected to HFD may result from increased oxidative stress and consequent excessive Ca^2+^ leak from the SR (through RyR1). Note that similar mechanisms also underlie MHS in humans, which is a disorder that shares many common features with HS. 

The take home message of our manuscript is that there are factors, in this case bad food habits, that may increase the risk of healthy people undergoing hyperthermic crises when exposed to high environmental temperature. 

The results collected in this study may have implications for the development of guidelines regarding food intake during heatwaves. Further studies should be carried out to investigate other risk factors and test the efficacy of treatments in preventing or blocking HS. However, it is also important to underline the possible limitations of our approach, one of them being the translation of the results obtained in mice to humankind. A first point to consider is the great variability of human genotype and lifestyle in contrast to the relative controlled condition of mice model. Moreover, our acute heat stress protocol (1 h at 41 °C) does not really reproduce the effect of a heat wave caused by the climate, which usually last several days (with lower temperatures). Longer exposure to elevated temperature could, in principle: (a) have a more severe effect on the physiology of skeletal muscle; or (b) give time to adaptation reducing the undesired effects. For this reason, we are already running a new set of experiments in which mice are exposed to 3 consecutive days of elevated temperatures (35 °C) to better mimic the classic heat waves caused by climate.

## Figures and Tables

**Figure 1 ijms-23-05286-f001:**
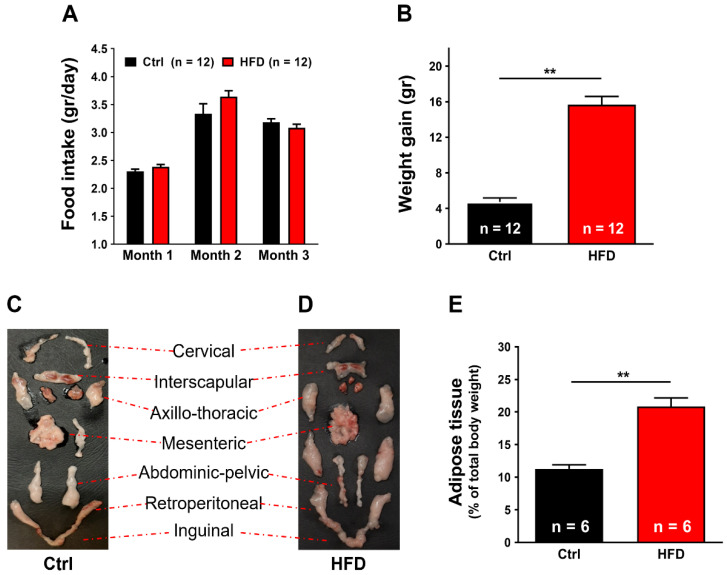
Food intake, weight gain, and body fat mass. (**A**) Food intake during the 3 months of HFD treatment (1–4 months of age). (**B**) Weight gain during the 3 months of HFD treatment. (**C**,**D**) Representative dissection of the total adipose tissue, from Ctrl (**D**) vs. HFD (**D**) mice. (**E**) Amount of adipose tissue expressed as percentage of total body weight. Data are shown as mean ± SEM (** *p* < 0.01, as evaluated by two-tailed unpaired Student’s *t*-test). *n* = number of mice tested.

**Figure 2 ijms-23-05286-f002:**
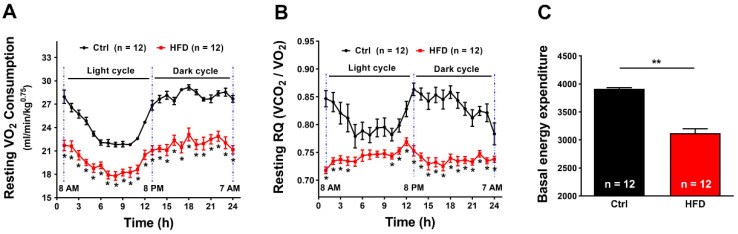
VO_2_ consumption, respiratory quotient (RQ), and basal energy expenditure during 24 h. (**A**) Oxygen consumption expressed as mL/min/kg^0.75^. (**B**) Respiratory quotient expressed as the ratio VCO_2_/VO_2_. (**C**) Area under the curve of 24 h-basal energy expenditure. Data are shown as mean ± SEM (* *p* < 0.05 and ** *p* < 0.01), as evaluated by two-way ANOVA followed by Tukey’s post-hoc test (panels (**A**,**B**)) and two-tailed unpaired Student’s *t*-test (panel (**C**)). *n* = number of mice tested.

**Figure 3 ijms-23-05286-f003:**
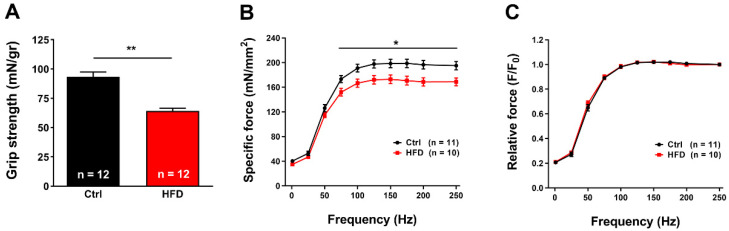
Grip strength and in-vitro specific force. (**A**) Grip strength normalized to body weight (in mN/gr). (**B**,**C**) Force frequency (1–250 Hz) curve of specific (panel (**B**)) and relative force normalized to 250 Hz (panel (**C**)) in isolated EDL muscles. Data are shown as mean ± SEM (* *p* < 0.05 and ** *p* < 0.01), as evaluated by two-way ANOVA followed by Tukey’s post-hoc test (panel (**B**)) or two-tailed unpaired Student’s *t*-test (panel **A**). *n* = number of mice tested (in panel (**A**)). *n* = number of EDL muscles tested (in panels (**B**) and (**C**)).

**Figure 4 ijms-23-05286-f004:**
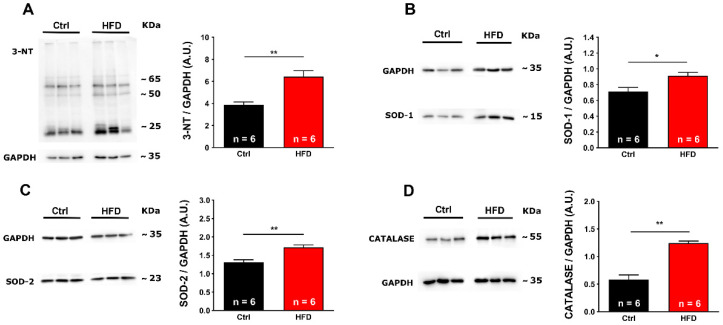
Markers of oxidative stress in EDL muscles. Representative immunoblots (**left**) and relative band densities normalized to GAPDH levels (**right**) of 3-NT (panel (**A**)), SOD-1 (panel (**B**)), SOD-2 (panel (**C**)), and Catalase (panel (**D**)) in EDL muscle homogenates. Data are shown as mean ± SEM (* *p* < 0.05 and ** *p* < 0.01), as evaluated by two-tailed unpaired Student’s *t*-test. *n* = number of EDL muscles.

**Figure 5 ijms-23-05286-f005:**
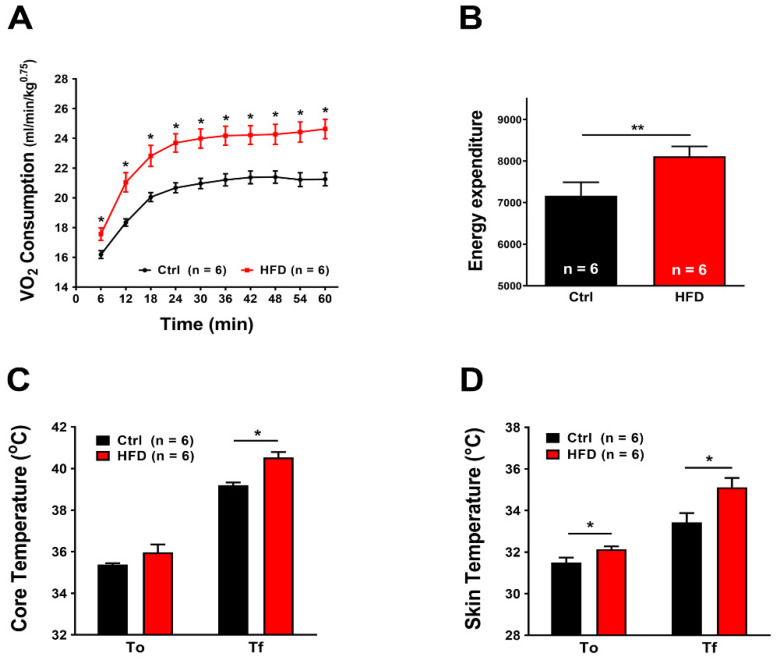
VO_2_ consumption, energy expenditure, and temperature recorded during heat stress protocol (41 °C for 1 h). (**A**) Oxygen consumption expressed as mL/min/kg^0.75^. (**B**) Area under the curve of 60 min energy expenditure. (**C**,**D**) Core and skin temperature, immediately before (T_0_) and at the end (T_f_) the heat stress protocol. Data are shown as mean ± SEM (* *p* < 0.05 and ** *p* < 0.01), as evaluated by two-way ANOVA followed by Tukey’s post-hoc test (panel **A**) or two-tailed unpaired Student’s *t*-test (panels (**B**–**D**)). *n* = number of mice tested.

**Figure 6 ijms-23-05286-f006:**
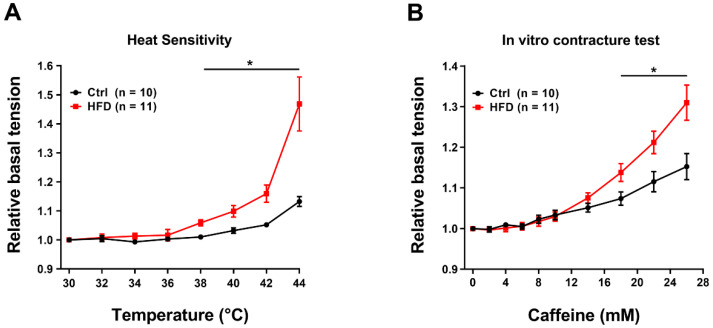
Temperature and caffeine dependence of basal tension in isolated EDL muscles. (**A**) Relative basal tension (normalized to 30 °C) during exposure to increasing temperature (from 30 to 44 °C). (**B**) Relative basal tension (normalized by 0 mM of caffeine) during exposure to increasing caffeine concentration (from 0 to 24 mM). Data are shown as mean ± SEM (* *p* < 0.05), as evaluated by two-way ANOVA followed by Tukey’s post-hoc test. *n* = number of EDL muscles tested.

**Figure 7 ijms-23-05286-f007:**
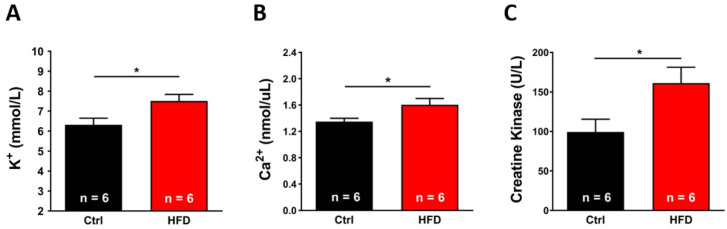
Blood markers of skeletal muscle damage after the heat stress protocol (41 °C for 1 h). Level of K^+^ in plasma (mmol/L) (panel (**A**)); level of Ca^2+^ in plasma (nmol/L) (panel (**B**)); amount of creatine-kinase (U/L) in plasma (panel (**C**)). Data are shown as mean ± SEM (* *p* < 0.05), as evaluated by two-tailed unpaired Student’s *t*-test. *n* = number of mice tested.

## Data Availability

Not applicable.

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
