# Peer review of "High-Fat Diet Impairs Muscle Function and Increases the Risk of Environmental Heatstroke in Mice"

_ijms, 2022, doi:10.3390/ijms23095286_

Round 1

Reviewer 1 Report

The investigators of the manuscript “High-fat diet impairs muscle function and increases the risk of 2 environmental heatstroke in mice” explored the underlying molecular mechanisms associated with heatstroke in a rodent model of HDF. The study's primary findings suggest that the 3-months of HFD may predispose rodents to HS susceptibility. This study points out that there are risks in bad food habits that might increase the chances of healthy people undergoing hyperthermic crises when exposed to high environmental temperatures. The study's observations address the urgent unmet need to draft guidelines on healthy food habits during heat waves. Overall, the study is very well planned and executed. However, to improve the quality of the current version of the manuscript, please include the following suggestion suggestions.

  1. Include the catalog no. of all the chemicals/ antibodies used in the study in the methodology section.
  2. Have a separate section to discuss the limitations of the current study

Reviewer 2 Report

The Authors of ijms-1684357 manuscript performed very interesting experiment on the influence of high-fat induced obesity on heatstroke. The whole stydy was planned properly and it complied all the requirements concernig animal experiments. Study protocol and all the methods used by the authors seem reliable. My only concern relates to the scope of investigated parameters. As the Authors caimed oxidative stress as a main reason of observed changes between high-fat and control groups it would be of utmost interest to check if the parameters related to protein oxidation (carbonyl groups) and lipid oxidation (TBARS, oxysterols) as well as total antioxidant capacity were changed.  These are the most common indicators of oxidative stress and they show the exact harmfull effects of oxidative stress in the organism. In my opinion it would enrich the manuscript. It is also important taking into account the fact, that the Authors cliam themselves, that decrease in force of muscles, which they observed in their study, may be caused by several factors, one of them being oxidative stress itself, as ROS and NOS are known to cause oxidative damage to proteins  and membranes and to alter their function. That is why it would be intersting to show the results related to oxidative damage of proteins and lipids.
